# Multiple Heat Source Thermal Modeling and Transient Analysis of LEDs

**Anton Alexeev** [1,*] **, Grigory Onushkin** [2] **, Jean-Paul Linnartz** [1,2] **and Genevieve Martin** [2]

[1] Department of Electrical Engineering, Eindhoven University of Technology, 5612 AZ Eindhoven, The Netherlands; j.p.linnartz@signify.com

[2] Research, Signify, High Tech Campus 7, 5656 AE Eindhoven, The Netherlands; grigory.onushkin@signify.com (G.O.); genevieve.martin@signify.com (G.M.)

* Correspondence: a.alexeev@tue.nl

**Abstract:** Thermal transient testing is widely used for LED characterization, derivation of compact models, and calibration of 3D finite element models. The traditional analysis of transient thermal measurements yields a thermal model for a single heat source. However, it appears that secondary heat sources are typically present in LED packages and significantly limit the model's precision. In this paper, we reveal inaccuracies of thermal transient measurements interpretation associated with the secondary heat sources related to the light trapped in an optical encapsulant and phosphor light conversion losses. We show that both have a significant impact on the transient response for mid-power LED packages. We present a novel methodology of a derivation and calibration of thermal models for LEDs with multiple heat sources. It can be applied not only to monochromatic LEDs but particularly also to LEDs with phosphor light conversion. The methodology enables a separate characterization of the primary pn junction thermal power source and the secondary heat sources in an LED package.

**Keywords:** dynamic thermal compact model; LED; silicone dome; phosphor light conversion; structure function; thermal transient analysis; thermal characterization; multiple heat source; secondary heat path

## 1. Introduction

The market of solid-state lighting has rapidly expanded over the past decades. Due to commoditization of light emitting diodes (LEDs), the expansion sets a need for accurate thermal modeling of LEDs to ensure highly reliable end products. Compact thermal modeling is an approach that enables fast and reliable thermal simulations of the devices of interest without disclosing confidential information from LED suppliers. Compact thermal models (CTMs) are widely used in modern production and in optimization processes (particularly in the semiconductor industry) [1–3]. One of the goals of our European Union project Delphi4LED [4,5] is the standardization of the CTMs for LEDs.

CTMs are traditionally either directly generated based on measurement data of thermal transients or are extracted by model order reduction [1] from detailed full-3D finite element thermal models [6]. CTMs derived from a 3D model can predict more thermal parameters of the original device. Yet a calibration of the original 3D model with thermal transient measurements is required. The calibration includes tuning of the thermal properties of the materials by the model's thermal transient response or structure function (SF) alignment with the data measured from the corresponding physical sample [7]. Geometry tuning and a proper choice of the right physical phenomena to be modeled should also be carefully executed.

The geometry of the 3D models can be extracted precisely from x-ray scans of a physical device [8]. The choice of the physics is complicated as multiple phenomena define the heat generation inside an LED package. Firstly, the LED chip generates heat due to internal optical losses (non-ideal internal quantum efficiency *IQE*), and electrical losses [9,10]. Secondly, the light emitted by the LEDs chip can be trapped in the encapsulating optical dome, partially re-absorbed inside the LED chip and on the walls of the LED package [11]. The impact of the heat loss caused by trapped light is often underestimated. We will show that these losses can have a significant influence on the LEDs transient and steady-state thermal behavior. Thirdly, if the LED has a silicone/phosphor layer, Stokes losses occur in the light conversion and cause thermal heating [12]. The phosphor material can cover the entire LED chip, be distantly located within the LED package, or be placed remotely on a secondary optic diffuser [13,14]. In the first or the second case, phosphor power losses can significantly affect the thermal performance of the LEDs [15].

Various approaches to calibrate multiple heat source LED thermal models are demonstrated in scientific works. A well-known impedance matrix approach enables the thermal characterization of multi-heat source multichip LEDs [16–18]. Yet this method cannot be applied for the characterization of the secondary heat sources: they are bounded to the main ones and cannot be activated separately. This makes the required measurements of transfer impedance impossible. The secondary heat sources can be characterized by ray tracing simulations [19–21]. Yet this approach requires access to proprietary information, such as phosphor composite particles spatial distribution, excitation and emission spectra, reflection properties, and the angular light distribution of the LED die. Another way to characterize secondary heat sources is comparative analysis of the thermal transient and optical measurements of an original bare chip LED and the same LED with a dispensed dome [13,22]. Yet, this method requires access to the bare chip LED packages, which are typically not available on the market and phosphor/silicone mixtures used by the LED manufacturers. Thus, we lack methods to characterize the LEDs secondary heat sources [23]. As a result, the trapped light and phosphor light conversion losses are typically coarsely approximated during the calibration of 3D models, e.g., Bornoff et al. assume 75% phosphor conversion efficiency without direct measurements when demonstrating their calibration procedure [7]. However, we show that the secondary heat sources have a significant impact on thermal transient and, therefore, these sources have to be estimated based on measurement data for proper calibration of 3D models. This work is filling the methodological gap by describing a procedure of multiple heat source LED thermal model calibration by analysis of LED package transient response.

We begin with familiarizing the reader with the considered LED package architecture, the transient analysis method, and the thermal SF concept. Next, we describe the LED's finite element analysis (FEA) thermal model. Then, we demonstrate an analytical estimation of the impact of such LED parameters as internal quantum efficiency (*IQE*), external quantum efficiency (*EQE*), light extraction efficiency (*LEE*), and dome geometry on the dome light extraction losses. Afterwards, we propose the novel thermal transient analysis methodology for multiple heat sources LED characterization. Then, we validate the proposed methodology and demonstrate the impact of the secondary heat sources on the interpretation of the transient measurements with our FEA model. We experimentally demonstrate the importance of the secondary thermal sources consideration for the thermal transient analysis of LEDs. Finally, we discuss the topologies of thermal resistor networks for physical-based modeling an LED. A list of abbreviations and variables can be found in the end of the paper.

## 2. Materials and Methods

### 2.1. LEDs Architecture

In this section, we give an overview of the LED packages of interest, their structures, characteristic dimensions and typical materials. We focus on mid-power (MP) surface mounted packages with lateral LED chips. MP LEDs are currently the most commonly used type in the lighting industry. Figure 1 represents a sketch of a typical MP LED package mounted on a metal core printed circuit

board (MCPCB). MP LEDs typically contain one to three separate dies interconnected with wire bonds. Each die is attached to a thermal pad by a die attach layer (DAL). For sake of simplicity, we consider a one-die LED package. Multichip LED packages with a larger number of dies can also be characterized by the approach presented in this paper. As shown in [24–26], the thermal transient measurements can also be applied to such packages. Yet, the interpretation of the results for CTM calibration must be done with careful approach. Numerous pn junctions in such packages may have different thermal behavior.

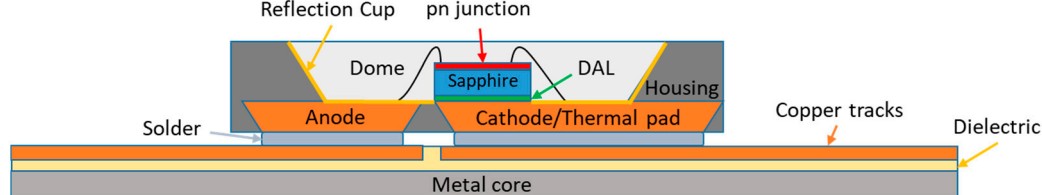

**Figure 1.** Sketch of a typical MP LED package soldered on an MCPCB.

A typical MP LEDs with square size of 3 mm × 3 mm (3030 package) have a characteristic reflection cup with a radius of approximately 1.2 mm. The representative height and width of the die are chosen as 0.2 mm and 0.8 mm, respectively. Highly efficient GaN LEDs are grown on c-plane sapphire substrate [27]. Despite the fact that sapphire crystal's thermal conductivity is anisotropic, the known crystal orientation enables high accuracy of sapphire thermal parameter estimation. The encapsulating dome above an LED die is typically fabricated out of either transparent silicone or silicone mixed with phosphor particles. LEDs' anodes, cathodes, and the thermal pads are traditionally made of copper.

### 2.2. Thermal Transient Analysis

A well-established method to characterize and extract a CTM of an LED is thermal transient analysis [28–30]. It requires sampling of device under test (DUT) junction temperature $T_j$ transient response data to an applied thermal power step $P_h$. The power step $P_h$ is derived as a difference between the measured supplied electrical power $P_{el}$ and the measured emitted optical power $P_{opt}$:

$$P_h = P_{el} - P_{opt} \tag{1}$$

A one-dimensional thermal RC Cauer network with a step response identical to $T_j$ is synthesized next. If the DUT heat path is sufficiently one-dimensional, the derived RC Cauer network genially represents its thermal properties. The details of the LED's transient characterization are described in the JESD51-5x series of standards [31–35].

A thermal RC Cauer network can be graphically represented as an SF. An SF is a plot of the cumulative $R_{th}$ values versus the cumulative $C_{th}$ values along the DUT heat path starting from the pn junction. An SF can be converted to a differential SF by taking the derivative of $C_{th}$ with respect to $R_{th}$. The peaks of differential SFs usually indicate new regions of the heat flow path [36]. We relate the thermal structures of the LED presented in Figure 1 to the particular regions of the correspondent SF and differential SF plots shown in Figure 2 according to the methods demonstrated in [28,37]. The pn junction is always located at the beginning of the plot. Thus, the initial increase of the thermal capacitance is related to the first element of the thermal path, the sapphire crystal. The subsequent shelf represents the DAL that has high partial $R_{th}$ and low partial $C_{th}$. A sharp increase of the cumulative thermal capacitance observed after the DAL shelf is related to the massive copper thermal pad. The thermal pad is attached to the copper tracks of the printed circuit board (PCB), a region with significant radial heat spreading. The SF image of it looks like a tilted straight line [28]. We use the differential SF peak related to the thermal pad to characterize the total $R_{th}$ of the die and the DAL. The peak is marked with a red arrow in Figure 2. Nevertheless, as will be shown further, the die and the DAL SF regions can be significantly distorted if the secondary heat sources are present. Thus, the $R_{th}$ value estimated with this method may be inaccurate. We analyze the factors contributing to the

SF distortion, the related physics and the impact of the secondary heat sources on thermal transient processes in this work. We use simulated SFs to quantify the impact.

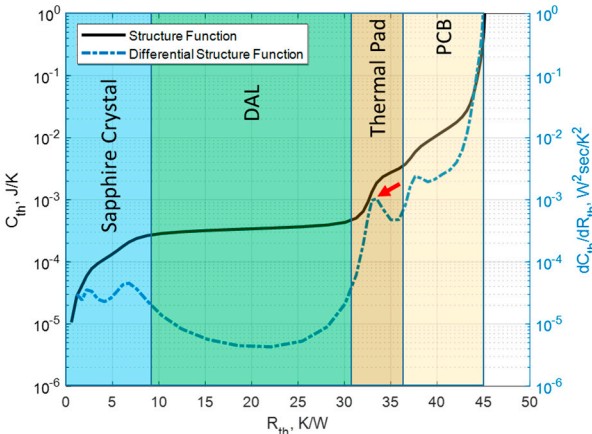

**Figure 2.** A characteristic structure function and a correspondent differential structure function of a typical one die MP LED and its relation to the LED's thermal heat path structures. The peak of the differential structure function related to the thermal pad is indicated.

The DAL is the most significant contributor to total $R_{th}$ of LED packages [38,39]. The partial $R_{th}$ of a DAL can reach up to tens of K/W for a single die MP LEDs package (Figure 2), while the partial $R_{th}$ values of other heat path structures as sapphire crystal and thermal pad are below 10 K/W. Thus, estimation of DAL properties is crucial for LEDs reliability prediction and calibration of FEA thermal models. The $R_{th}$ of a DAL is the major contributor to junction to thermal pad thermal resistance $R_{th\_J2T}$. For our further analysis of the SF distortion we compare the $R_{th\_J2T}$ values derived with two different methods: SF analysis and FEA. The correspondent values are named $R_{th\_J2T\_SF}$ and $R_{th\_J2T\_FEA}$, respectively.

As shown before, the location of the correspondent differential SF peak enables a straight-forward identification of the $R_{th\_J2T\_SF}$ value. Yet, the peak might become blurred and impossible to identify by this method for some LEDs configurations. In this case the $R_{th\_J2T\_SF}$ value is determined by a method inspired by [40] as cumulative $R_{th}$ value correspondent to characteristic $C_{th\_J2T}$ value which is defined as:

$$C_{th\_J2T} = C_{th\_crystal} + 0.5C_{th\_thermal\ pad} \qquad (2)$$

$R_{th\_J2T\_SF}$ values derived by these methods are in excellent agreement.

We use $R_{th\_J2T\_FEA}$ to verify $R_{th\_J2T\_SF}$ values and determine the accuracy of the thermal transient measurements interpretation. The $R_{th\_J2T\_FEA}$ derivation method is based on a direct evaluation of steady-state thermal FEA results. It requires calculation of $\overline{T_j}$ and $\overline{T_{DAL}}$, the average temperatures of the pn junction and the bottom of the DAL finite element model nodes, respectively. According to the definition of the thermal resistance:

$$R_{th\_J2T\_FEA} = \frac{\overline{T_j} - \overline{T_{DAL}}}{P_{hJ}} \qquad (3)$$

The difference between the $R_{th\_J2T\_SF}$ and the $R_{th\_J2T\_FEA}$ defines the accuracy of SF representation of the LED's main heat path. We define the relative error of $R_{th\_J2T\_SF}$ derivation as:

$$R_{th\_J2T\_err} = \frac{R_{th\_J2T\_FEA} - R_{th\_J2T\_SF}}{R_{th\_J2T\_FEA}} \qquad (4)$$

### 2.3. Thermal FEA Modeling

We use a modification of the FEA model demonstrated in our previous publication [41]. The model was built with the MATLAB (2018b, MathWorks, Natick, MA, USA) partial differential equation toolbox. We aim to determine the effects of the secondary heat sources on the accuracy of the transient analysis, in particular, on the $R_{th\_J2T}$ value. We use a generalized axisymmetric geometry of an LED package mockup. Our model contains only the major thermal elements of an LED: pn junction, sapphire crystal, DAL, thermal pad, and dome. The fine details of the LED package are omitted since their effect on the $T_j$ transient response and SF is negligible and unique for each LED package. The model can simulate characteristic thermal behavior of MP LEDs packages of various size by adjusting the characteristic dimensions of the thermal structures. Figure 3a depicts the geometry of the FEA thermal model and defines key characteristic dimensions. The geometrical and thermal parameters of the model are presented in Table 1.

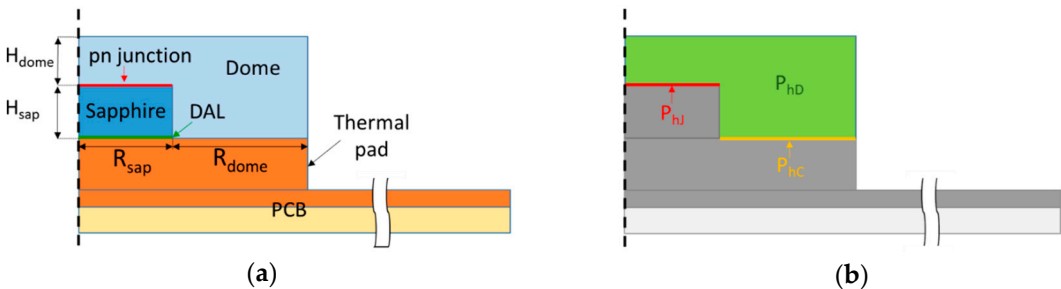

**Figure 3.** (**a**) Cross-sectional view of the axisymmetric FEA LED package mockup and its dimensions. (**b**) Thermal power dissipation regions.

**Table 1.** Geometrical and thermal properties of the LEDs FEA model structures.

| Element | Material | Radius (mm) | Height (mm) | $\rho$ (g/mm$^3$) | $C$ (J/gK) | $k$ (W/mK) |
|---|---|---|---|---|---|---|
| Crystal | Sapphire | 0.4 | 0.2 | 3.98 | 0.85 | 32 |
| DAL | - | 0.4 | 0.002 | - | - | 0.1 |
| Thermal pad | Copper | 1.2 | 0.25 | 8.93 | 0.39 | 380 |
| Dome | Silicone phosphor | 0.8 | 0.2 | 1.10* | 1.15* | 0.2* |
| MCPCB tracks | Copper | 10 | 0.07 | 8.93 | 0.39 | 380 |
| MCPCB dielectric | FR4 | 10 | 0.1 | 1.90 | 1.2 | 0.2 |

* The data are given for silicone. Silicone/phosphor composite thermal properties are defined in Equations (22)–(24).

Plastic housing of LEDs is not explicitly considered in the thermal simulations because its thermal conductivity is considerably lower than that of the copper thermal pad. Moreover, the housing is separated from the junction by a dome made of an extremely low thermal conductive material. Thus, we assume the housing thermal effect on the $T_j$ response to be small. Nevertheless, as shown in [42], the housing geometry significantly affects the *LEE* and the trapped light thermal losses. The trapped light thermal losses related to the plastic housing walls (cup reflector surface) are added to the thermal losses on top of the thermal pad surface.

The model contains three heat sources. The sources are defined in Figure 3b and related to the pn junction $P_{hJ}$, dome $P_{hD}$, and cup reflector surface $P_{hC}$. $P_{hJ}$ and $P_{hC}$ are homogeneously distributed over the corresponding junction and cup surfaces. $P_{hD}$ is uniformly distributed within the dome volume. The sum of the considered heat sources is equal to the total thermal power losses $P_h$ of the LED:

$$P_h = P_{hJ} + P_{hD} + P_{hC} \tag{5}$$

We define the sum of the parasitic secondary heat losses as $P'_h$, with:

$$P'_h = P_{hD} + P_{hC} \tag{6}$$

We also evaluate heat transfer through the copper tracks and the dielectric layer of the MCPCB in our model. A constant temperature boundary condition is set at the bottom of the PCB's dielectric layer to simulate the upper surface of the PCB's highly conductive metal core. We set uniform initial temperature condition. Convection and radiation heat transfers from the outer surfaces of the LED package are ignored. A number of works has previously demonstrated that a negligible fraction of heat leaves an LED via these mechanisms compared to conduction via the main heat path [43,44]. Thus, adiabatic boundary conditions are applied to all outer LED surfaces. Therefore, the numerical problem is linear and the system temperature response, normalized with applied thermal power, does not depend neither on the absolute value of the initial temperature nor on the applied power. The heat sources are activated at $\tau = 0$. Transient response of the averaged pn junction temperature is evaluated. For calculation of the $R_{th\_J2T\_FEA}$ the steady-state averaged temperature of the bottom of the DAL finite element model nodes is used.

### 2.4. Experimental Setup and Physical Specimens

To confirm the impact of secondary heat sources, experiments using two samples of MP 3030 LED with a bare non-encapsulated chip have been conducted. We made various modifications for the dome encapsulant of each LED. The 1st LED got a flat silicone dome as an encapsulant filling the package cup. The 2nd LED was initially modified with a clear silicone rim layer around the sapphire chip and then a silicone/phosphor mixture as a light conversion layer covering the whole open surface in the package cup. OE-6650 resin material (Dow Corning, Midland, Michigan, United States) was used as the silicone for these layers. The resulting LED configurations are presented in Figure 4. Transient testing of each of them was performed separately. Comparative analysis of the results correspondent to each LED enabled identification of the impact of the trapped light and the phosphor secondary power sources.

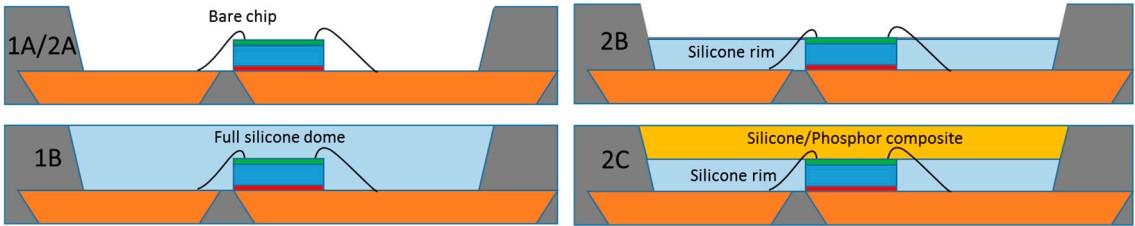

**Figure 4.** The physical LED specimens' configurations.

To perform the thermal transient measurements, the LEDs were soldered on an MCPCB which was mounted on a heat sink. T3Ster (Mentor Graphics, Wilsonville, Oregon, United States) thermal tester was used to preform transient testing. An on/off forward voltage response was measured while the heat sink temperature was kept 50 °C. We used 150 mA driving and 10 mA measurement currents. The dependence of the LEDs' forward voltage on $T_j$ was calibrated in the range of 25–85 °C while the LEDs were mounted on a coldplate and biased with the measurement current. We use a quadratic approximation of the $T_j$ dependence on the forward voltage to increase the accuracy, since it was shown in the literature that use of a linear approximation may lead to a significant error of the junction temperature evaluation for LEDs [45,46].

We used a square root initial correction to substitute the initial electrical transient. The correction was based on $T_j$ response data sampled in the 50–400 µsec interval. The optical fluxes were measured by an integration sphere to calculate the total thermal power $P_h$ dissipated by each LED. The LEDs were horizontally mounted on a heat sink during the optical measurements in order to minimize the convention heat flux losses that may be caused by the non-isothermal environmental conditions at open interface with integration sphere [45]. The LEDs' forward current and the heat sink temperature were kept the same during all thermal transient measurements.

### 2.5. Multiple Heat Source Characterization

In this chapter, we perform an analysis of the fraction of the secondary thermal power sources in the total power dissipation for MP LED packages. First, we analytically estimate the power losses related to the trapped light $P_{hC}$. We demonstrate a method to estimate $P_{hC}$ based on the values of *IQE*, *EQE*, and *LEE* parameters and on the geometries of the LED chip and the package. Next, we estimate the thermal losses caused by phosphor light conversion $P_{hD}$. Finally, we propose an experimental method that enables separation of the secondary heat sources power $P'_h$ from the total thermal power $P_h$. The method is based on a revised analytical solution of an initial thermal transient response. We analyze the applicability of the proposed method for LEDs with transparent domes and for LEDs with phosphor light conversion.

### 2.5.1. Secondary Heat Sources

In this subchapter, we aim to estimate the fraction that secondary heat sources contributing to the total heat dissipation. First, we analyze LEDs without phosphor light conversion. In this case the secondary heat sources are originated from the light trapped in the encapsulating dome layer due to total inner reflections (TIRs), (see Figure 5). This light is partially absorbed by the reflector cup walls, the reflective metal contact pads, and the pn junction when light re-enters the LED crystal [47].

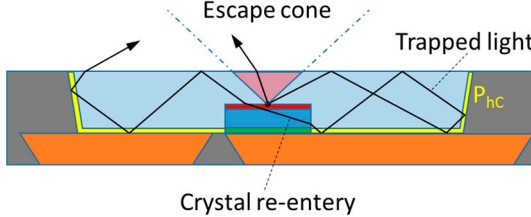

**Figure 5.** Illustration of a trapped light ray emitted by the pn junction experiencing multiple TIRs before leaving a flat silicone dome of an LED.

In general, evaluation of the trapped light losses requires sophisticated ray tracing modeling and extensive LED characterization. Yet, it is possible to make a coarse analytical estimation employing approximations of the *LEE* and *IQE* coefficients and other LED parameters, such as cup reflectivity, dome curvature, die and package dimensions, etc. First, we define the total thermal power $P_h$ dependence on the applied electrical power. The total *LEE* of an LED package is a product of the chip-to-dome light extraction coefficient $LEE_{chip}$ and the dome-to-air light extraction coefficient $LEE_{dome}$, thusly:

$$LEE = LEE_{chip} \cdot LEE_{dome} \tag{7}$$

*EQE* is a product of *IQE* and *LEE*, that is:

$$EQE = IQE(I) \cdot LEE_{chip} \cdot LEE_{dome} \tag{8}$$

Unlike $LEE_{chip}$ and $LEE_{dome}$, *IQE* is dependent on the forward current *I*. The total thermal power dissipated in an LED is:

$$P_h = P_{el} \cdot \left(1 - IQE(I) \cdot LEE_{chip} \cdot LEE_{dome}\right) \tag{9}$$

The fraction of the light initially left the die and trapped on the reflection cup walls is:

$$P_{hC} \cong IQE(I) \cdot LEE_{chip} \cdot (1 - LEE_{dome}) \cdot \Lambda \tag{10}$$

where coefficient $\Lambda$ provides a correction for the crystal light re-absorption. Appendix A analyzes $\Lambda$ dependence on the LED's package geometry and other parameters.

The $LEE_{dome}$ coefficient is difficult to measure directly without manufacturing packages of custom calibration LEDs and without advanced measurement setups. Therefore, we use the results of ray tracing simulation presented by Tran et al. [42]. The results enable estimation of the $LEE_{dome}$ for MP LED packages. The authors define the $LEE_{dome}$ of an LED, dependence on the dome curvature, and angle of the reflector cup. The simulation results evidence that the $LEE_{dome}$ coefficient for conventional dome designs varies from 0.65–0.92. The $LEE_{dome}$ coefficient for conventional multiple-chip LEDs with flat light-emitting surface (LES) and the absence of special light extraction enhancement structures is around 60% [48–50]. Optimization of the LEDs packages by using a gradient refractive index encapsulant, roughened or patterned lead-frame substrates, and the scattering effect of phosphor particles can increase the $LEE_{dome}$ up to 85%.

Now we estimate the fraction of $P_{hC}$ in the total power dissipated by an LED:

$$\frac{P_{hC}}{P_h} \cong \frac{IQE(I) \cdot LEE_{chip} \cdot (1 - LEE_{dome})}{\left(1 - IQE(I) \cdot LEE_{chip} \cdot LEE_{dome}\right)} \cdot \Lambda \qquad (11)$$

We then estimate the $P_{hC}/P_h$ ratio for blue GaN LEDs. Modern state-of-the-art high brightness blue LEDs can have a $LEE_{chip}$ coefficient of 85% for double-side textured-crystals [51]. The reflectance coefficient of the cup is set to 93%. The ratio between the areas of the cup and the crystal $S_{cup}/S_{cry}$ plays an important role. It defines the probability of crystals re-entry by the trapped light. We consider two $S_{cup}/S_{cry}$ ratios: 7 and 2. The first corresponds to the MP LED architecture presented in Table 1. The second represents a smaller single die or a multiple die LED package. Indeed, multiple-die packages with low $S_{cup}/S_{cry}$ ratios have increased shielding of the light by the neighboring dies (denser chip placement leads to an increased light re-absorption by the LED package chips) [52]. The results are presented in Figure 6.

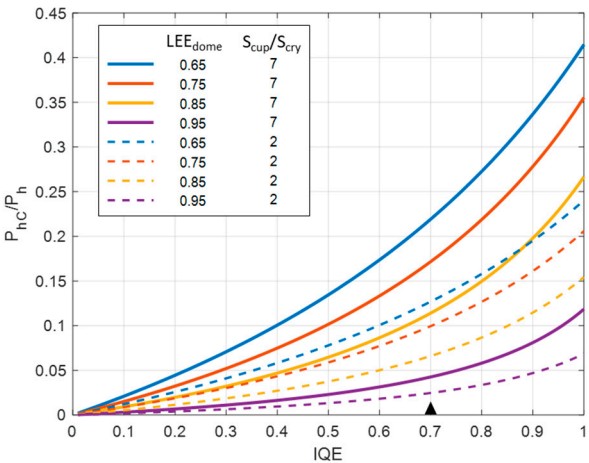

**Figure 6.** $P_{hC}/P_h$ ratio as a function of *IQE* for a range of $LEE_{dome}$ coefficient.

Figure 6 plots evidence that LED chips with high *IQE* in low $LEE_{dome}$ packages have the highest relative $P_{hC}$ thermal losses. The fraction of $P_{hC}$ rapidly increases with increase of *IQE*. A decrease of the $S_{cup}/S_{cry}$ value leads to a decrease of the $P_{hC}$ fraction due to the shielding effect.

The *IQE* of blue GaN LEDs approaches its theoretical limit of 95%. Nevertheless, in the majority of high-power applications, blue LEDs are driven in the droop regime when the *IQE* is approximately 70% (marked at the plot). These parameters indicate that approximately up to 25% of the total thermal power $P_h$ can be dissipated on the LED cup walls due to TIR.

LEDs with a phosphor light conversion layer always have extra heat losses due to Stokes effect. This emphasizes the fact that the secondary heat sources are significant. Next, we propose a methodology of their experimental estimation.

2.5.2. Estimation of the Secondary Heat Sources

Modern LED transient testing methods determine the total thermal power $P_h$ as a difference between the applied electrical power $P_{el}$ and the emitted radiant flux $P_{opt}$ [32–35]. It is impossible to separate the secondary heat sources $P'_h$ with this approach. In this section, we propose a method of experimentally estimate $P'_h$. The method is based on a revised solution of the initial $T_j$ transient response.

In practice, it is challenging to measure the initial $T_j$ response during the first tens of milliseconds due to the electrical transient processes in the pn junction, the connecting wires and the transient measurement equipment. Therefore, correction methods were developed to restore this data. One of these methods is a square root correction. The method is based on the analytical solution of heat propagation into a semi-infinite material from a homogeneous surface heat source: the pn junction substrate can be often approximated as a surface heat source and a semi-infinite body in the beginning of thermal transient. The initial $T_j$ transient response can be approximated with the following equation [53,54]:

$$T_j(t) = T_0 + \frac{P_{hJ}}{S_{pn}} K \sqrt{t} \tag{12}$$

The form of the equation evidences that the initial $T_j$ transience response plotted versus a square root of time is linear. The coefficient $K$ bounds the slope of this plot with the junction dissipated thermal power $P_{hJ}$ and the chip surface $S_{pn}$. An example of measured LED data and of the applied square root initial transient correction based on 50–400 μsec interval is shown in Figure 7.

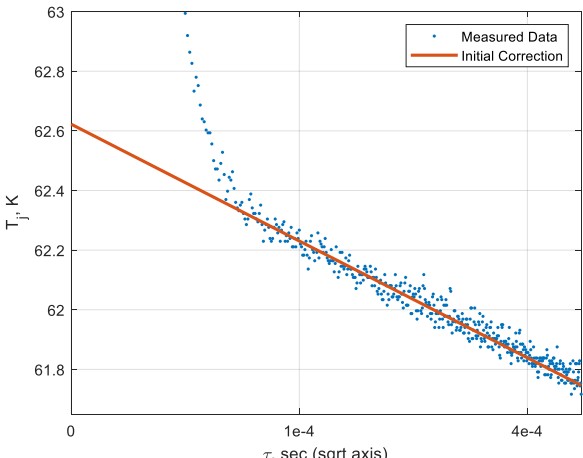

**Figure 7.** An example of the square root initial transient correction applied to measured data of an on/off transient.

The correction method assumes a unilateral one-dimensional heat propagation in the beginning of the transient. While LEDs chips are typically encapsulated in a dome to enhance light extraction and shape the light beam. Thus, the heat dissipated by the pn junction propagates bilaterally, both to the substrate and the dome. We solve a problem of the bilateral heat transfer into two semi-infinite bodies representing the sapphire crystal substrate and the encapsulating dome. We solve a problem of the bilateral heat transfer into two semi-infinite bodies representing the sapphire crystal substrate and the dome. The solution is presented in Appendix B. The resulting equation retains the square root time-dependence. After substituting the power density $q$ with $P_{hJ}/S_{pn}$ in Equation (A17) we obtain the coefficient $K_{bi\_lat}$ characterizing $T_j$ initial response for the case of bilateral heat propagation. The original coefficient $K_{uni\_lat}$ and the derived $K_{bi\_lat}$ are presented below:

$$K_{uni\_lat} = \frac{2}{\sqrt{\pi k_s \rho_s C_s}} \tag{13}$$

$$K_{bi\_lat} = \frac{2}{\sqrt{\pi}\left(\sqrt{k_s \rho_s C_s} + \sqrt{k_d \rho_d C_d}\right)} \tag{14}$$

The similar form of the bilateral heat propagation solution justifies the application of the square root initial correction for LEDs. Nevertheless, the bilateral solution bounds quite accurately the initial $T_j$ response to the thermal power dissipated by the junction $P_{hJ}$.

We determine the values of the heat flows toward the dome and the substrate during the initial thermal transient by substituting the correspondent thermal properties in Appendix B Equations (A15) and (A16). We derive that approximately 5% of the total heat is dissipated by the junction propagates to the dome. This corresponds to approximately 5% systematic error for $P_{hJ}$ power evaluation with the classic method when no heat propagation into the dome is considered.

The $T_j$ response follows the square root time dependency only for a finite amount of time while the assumption of one-dimensional heat propagation is valid on both sides of the active region. On the one hand, it is limited by the characteristic time constant of the substrate $\tau_s$. On the other hand, by the requirement of sufficiently one-dimensional heat propagation into the dome.

The substrate time constant $\tau_s$ can be found as a product of the partial thermal resistance and thermal capacitance of the die crystal, which can be expressed with sapphire thermal properties and the crystal height $H_{sap}$:

$$\tau_s = R_{th\_sap} C_{th\_sap} = \frac{H_{sap}^2 \rho_{sap} C_{sap}}{k_{sap}} \tag{15}$$

The heat propagation into the dome can be considered sufficiently one-dimensional if the characteristic width of the pn junction $l'_{pn}$ is much larger than the characteristic depth of heat propagation into the dome $x'_d$:

$$\frac{x'_d}{l'_{pn}} \ll 1 \tag{16}$$

We estimate $l'_{pn}$ as a one half of the minimal dimension of the top of the sapphire crystal. Carslaw and Jaeger [55] have derived a closed-form solution of the time-dependent temperature profile for heat propagation into a semi-infinite media:

$$T(x,t) = T_0 + \frac{2q}{\sqrt{\pi k \rho C_p}} \sqrt{t} \exp\left(-\frac{x^2}{4\tau}\frac{\rho C}{k}\right) - \frac{qx}{k} erfc\left(\sqrt{\frac{x^2}{4\tau}\frac{\rho C}{k}}\right) \tag{17}$$

The form of the exponential term of the Equation (17) yields the $x'_d$ dependence on time:

$$x'_d \approx 2\sqrt{\tau \frac{k_d}{\rho_d C_d}} \tag{18}$$

Thus, the characters time $\tau_d$ at which $x'_d \approx l'_{pn}$ is:

$$\tau_d \approx \frac{1}{4}\frac{{l'_{pn}}^2 \rho_d C_d}{k_d} \tag{19}$$

Therefore, the initial heat propagation is sufficiently one-dimensional on both sides of the active region if:

$$\tau \ll \min(\tau_s, \tau_d) \tag{20}$$

$\tau_s$ and $\tau_d$ are estimated as 3.8 msec and 250 msec, respectively, considering the data of Table 1. Estimated value of $\tau_s$ is significantly smaller than $\tau_d$. Thus, we conclude that the heat propagation into the silicone dome is always sufficiently one dimensional until the heat flux has not reached the DAL via sapphire substrate. Therefore, $P_{hJ}$ can be reliably extracted by Equations (12) and (14). Then, $P'_h$ can be found as a difference between $P_h$ and $P_{hJ}$.

### 2.5.3. Applicability of the Approach for LEDs with Phosphor Light Conversion

In this section, we investigate the applicability of the proposed method of $P_{hJ}$ estimation for LEDs with silicone/phosphor composite domes. The heat generation by phosphor particles may disturb the initial thermal transient which can affect the accuracy. We investigate the impact of the presence of phosphor on the accuracy of the proposed $P_{hJ}$ extraction method.

Firstly, we perform estimation of the thermal properties of the phosphor/silicone composite material. We use the phosphor filler volume fraction $f$ parameter defined with the volumes of the silicone $V_{sil}$ and the phosphor $V_{pho}$ fractions as:

$$f = \frac{V_{pho}}{V_{pho} + V_{sil}} \tag{21}$$

Numerous models have been proposed to model the effective thermal conductivity of this type of composites [56–58]. These models are typically derived for a certain range of the phosphor volume fraction. We use a high volume fraction limit model proposed by Every [59]. The effective thermal conductivity $k_d$ of the silicone/phosphor composite material dome is expressed as:

$$k_d = \frac{k_{sil}}{(1-f)^{3(1-\alpha)(1+2\alpha)}}. \tag{22}$$

Here, $k_{sil}$ is the thermal conductivity of the silicone and $\alpha$ is a nondimensional parameter bounding the particle size and particle-composite matrix interface effect.

Zhang et al. [60] have fitted the model of the equation to experimental measurements of the typical silicone/phosphor composite used in LEDs They used and $Ce^{3+}$ doped YAG ($Y_3Al_5O_{12}$) phosphor particles of 13.0 ± 2.0 μm diameter encapsulated in high optical transparency silicone. They achieved an excellent agreement with the experimental results for high volume concentrations $f$ from 20–40%. It was found that $\alpha$ is 0.004. This fitting result slightly overestimate the thermal conductivity for the composites with low phosphor volume fractions.

The YAG phosphor density $\rho_{pho}$ and the specific heat $C_{pho}$ are 4.56 g/cm$^3$ and 0.6 J/(g K), respectively [61]. The density $\rho_d$ and specific heat $C_d$ of the dome composite are estimated based on the volume fraction of the phosphor particles $f$ as:

$$\rho_d = \rho_{sil}(1-f) + \rho_{pho}f \tag{23}$$

$$C_d = C_{sil}(1-f) + C_{pho}f \tag{24}$$

The phosphor dome light conversion efficiency was shown to be dependent on the phosphor particles type, concentration, temperature, experienced thermal stress [13,44,62,63]. Yet, we assume the silicone/phosphor composite properties to be constant during the fast initial transient processes due to small temperature variations. Consequently, the heat transfer problem remains linear. Therefore, we address the multiple heat source initial thermal transient analysis using the principle of superposition. We analyze heating of the LED package with $P_{hD}$ and $P_{hJ}$ heat sources separately and compare the heating rates.

We start with the analysis of the silicone/phosphor composite dome. The data presented by Chung [64] shows the evidence that the heating time constant of the remote phosphor layers is significantly slower than the pn junction heating time constant (1 min vs. 0.02 sec). However, the phosphor layers deposited over the pn junction may have significant higher heating rates due to the higher optical power density. Lou et al. [20,65] determined the phosphor energy conversion efficiency both experimentally and numerically. For warm white LEDs with high phosphor volume fractions, up to 45% of blue light optical power can be dissipated as heat during light conversion [66]. If we assume that $WPE_{chip} = 70\%$ for the LED chip (a typical value for modern blue LEDs under typical

operational conditions), then it will mean that up to half of the total thermal power $P_h$ for white LED package can be related to the phosphor thermal losses $P_{hD}$. Similar power ratio results were previously determined in the literature [19,22].

The rate at which the temperature of the silicone/phosphor composite dome increases during the first hundreds of milliseconds after turning an LED on is linear. It is determined mainly by the capacitive thermal effects due to the low thermal conductivity of the silicone/phosphor composite. Thus, we estimate the composite dome temperature increase $\Delta T_d$ at times $\tau \ll \tau_d$ as:

$$\Delta T_d(\tau) = \frac{P_{hD}\tau}{\rho_d C_d V_d} \tag{25}$$

Here $V_d$ is the characteristic dome volume per die. Multiple die LEDs have higher dome optical power density than the one-die LEDs. This leads to a faster rate of silicone/phosphor composite heating. To consider a worst-case scenario we perform an estimation for a case of two-die MP 3030 white LED package. Thus, we chose $V_d$ as a half of the correspondent LED dome volume.

We compare the characteristic dome temperature increase with the junction temperature increase $\Delta T_j(\tau)$ estimated by Equation (12) and $K_{bi\_lat}$ coefficient. The comparison yields evidence that for $\tau$ below 400 µsec (which is a typical upper time limit used for transient correction), $\Delta T_d$ is less than $0.03 \cdot \Delta T_j$. Thus, the initial heating rate of the silicone/phosphor composite is significantly slower than one of the pn junction. Moreover, the thermal conductivity of the silicone/phosphor composite is considerably lower than one of the sapphire crystal. All these factors indicate that the phosphor-related thermal losses have insignificant impact on the initial transients, as will be confirmed in the next chapter.

The unilateral coefficient $K_{uni\_lat}$ and the derived dependence of the bilateral coefficient $K_{bi\_lat}$ as a function of the phosphor fraction $f$ are shown in Figure 8. The data confirms 5–15% $K_{uni\_lat}$ relative error if compared with more precise $K_{bi\_lat}$ values. The same error will have $P_{hJ}$ when estimated by Equation (12) under the classical unilateral heat propagation assumption.

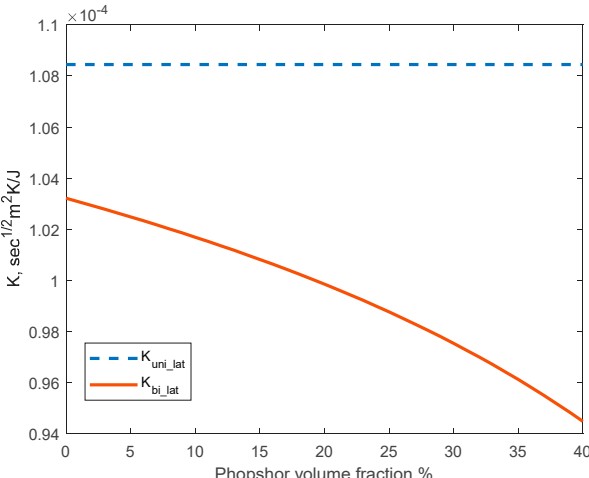

**Figure 8.** Comparison of $K_{uni\_lat}$ and $K_{bi\_lat}$ coefficients.

## 3. Results

### 3.1. Numerical Simulations

In this section, we determine the impact of the power $P'_h$ for secondary heat sources on the transient analysis results with our FEA model. The analytical estimations presented above show evidence that $P_{hC}$ can reach up to 25% of total $P_h$ for LEDs with inefficient dome design. Phosphor-related thermal losses $P_{hD}$ can be estimated as high as 50% of $P_h$ if we consider an LED that has 65% phosphor light conversion efficiency [14] driven with current correspondent to 75% *IQE*. We aim to determine

general patterns and estimate the thermal transient analysis inaccuracies associated with the secondary heat sources.

### 3.1.1. Secondary Heat Sources Impact on Junction Thermal Transient

First, we analyze the logarithmic time derivative and the initial transient of $T_j$ response of the FEA model (Figure 9). We consider two extreme cases: $P'_h = P_{hD}$ and $P'_h = P_{hC}$.

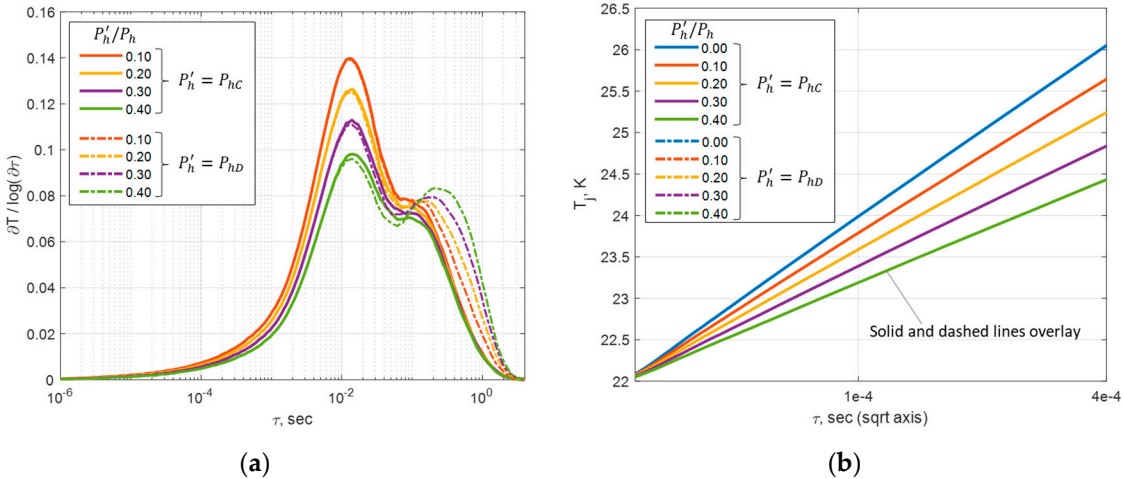

(a)                                          (b)

**Figure 9.** (a) Log time derivative of $T_j$ response. The solid lines represent LEDs with $P'_h = P_{hC}$. The dashed lines are correspondent to LEDs with $P'_h = P_{hD}$. (b) Initial $T_j$ response vs. The square root of time. No considerable difference between the $P'_h = P_{hD}$ and $P'_h = P_{hC}$ cases is observed, the dashed lines are merged with the solid ones.

Figure 9a shows that the initial thermal responses are similar up to 10 msec. They linearly scale with the $P'_h/P_h$ fraction. A significant effect of the phosphor heat generation is observed for $\tau > 10$ msec. Figure 9b shows that the initial $T_j$ transient responses plotted versus square root of time are linear up to 400μsec. As estimated in Section 2.5.3, phosphor thermal power generation $P_{hD}$ has no influence on the initial transient in the considered time interval. Thus, the numerical modeling confirms that the $K_{bi\_lat}$ coefficient extracted from the initial thermal transient is weakly affected by the heat generation in the encapsulating dome. This confirms the applicability of the $P_{hJ}$ estimation from initial transient response analysis even for LEDs with phosphor light conversion layer.

### 3.1.2. Secondary Heat Sources Estimation Verification

To verify the proposed method of $P'_h$ and $P_{hJ}$ evaluation, we determine the increase of the $\Delta T_j$ at $\tau = 400$ μsec by deriving $K_{bi\_lat}$ and $K_{uni\_lat}$ coefficients using Equations (12)–(14). We compare the obtained $\Delta T_j$ values against the simulation results in Table 2. The data show that the proposed bilateral initial heat propagation model predicts initial thermal transient response with significantly higher accuracy as compared to the classical unilateral model. Thus, the $K_{bi\_lat}$ coefficient helps to decrease significantly the $P'_h$ and $P_{hJ}$ evaluation errors.

**Table 2.** Initial $\Delta T_j$ response predicted by unilateral and bilateral analytical models at $\tau = 400$ μsec.

| Phosphor Volume Fraction | $\Delta T_j$ (K) | | |
|---|---|---|---|
| | FEA Reference | Bilateral Estimation | Unilateral Estimation |
| 0 | 4.1 | 4.1 | 4.3 |
| 0.4 | 3.7 | 3.7 | 4.3 |

### 3.1.3. Secondary Heat Sources Impact on Structure Functions

For numerical simulations of the LED thermal structure functions we consider two extreme cases: $P'_h = P_{hC}$ and $P'_h = P_{hD}$. We change the secondary heat sources $P'_h/P_h$ fraction from 0 to 0.4. We use $P_h$ as an input power for thermal transient analysis. The resulting simulated SFs are presented in Figure 10a. SF-derived $R_{th\_J2T\_SF}$ and reference $R_{th\_J2T\_FEA}$ thermal resistances are indicated on the graph. The $R_{th\_J2T\_FEA}$ is found to have an extremely weak dependence on the secondary heat sources presence, thus, $R_{th\_J2T\_FEA}$ is considered to be independent of the secondary heat source's power fraction. The blue SF represents the LED without secondary heat sources. The DAL SF based estimation error $R_{th\_J2T\_err}$ defined by Equation (4) is plotted in Figure 10b.

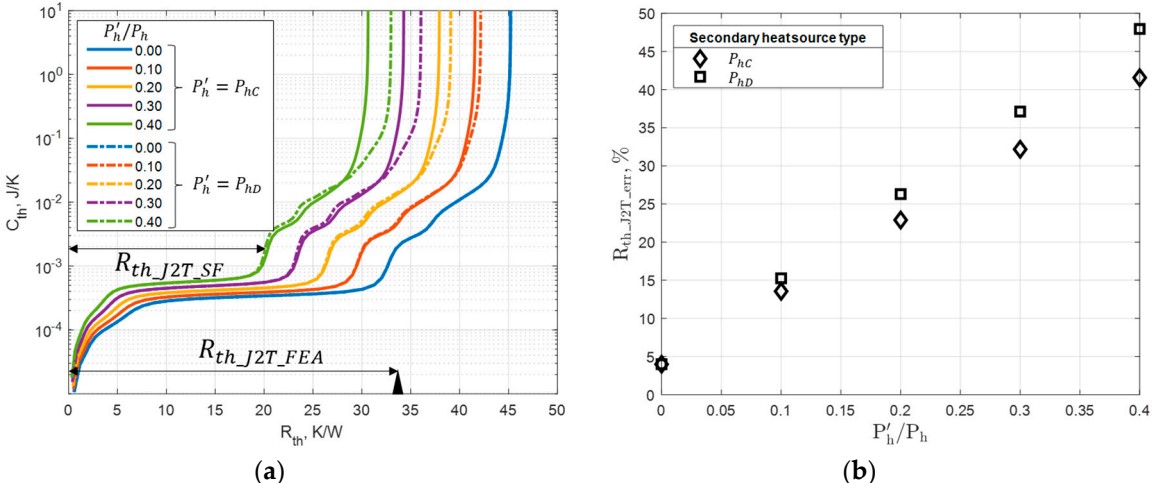

**Figure 10.** (**a**) Influence of the secondary heat sources on SFs. The solid structure functions represent LEDs with a heat source imitating the dome trapped light losses. The dashed structure functions are related to LEDs with heat source imitating the phosphor light conversion losses. The reference thermal resistance $R_{th\_J2T\_FEA}$ is indicated. (**b**) Error of the $R_{th\_J2T\_SF}$ part relative to the $R_{th\_J2T\_FEA}$.

We notice that the error is approximately linearly proportional to the $P'_h/P_h$ ratio for both considered cases:

$$R_{th\_J2T\_err} \simeq P'_h/P_h \tag{26}$$

Any actual configuration of secondary thermal sources within an LED is formed as a superposition of these extreme cases. Since the considered problem is linear, thus, our proposed $R_{th\_J2T\_err}$ estimation is valid for both types of encapsulated LEDs, with and without phosphor light conversion.

$R_{th\_J2T\_err}$ is slightly higher for the $P'_h = P_{hD}$ than for $P'_h = P_{hC}$ case. The difference is explained by the fact that in steady-state conditions a fraction of the heat generated by the phosphor containing encapsulating dome goes through the sapphire crystal and through the DAL, while the heat generated by the trapped light in clear encapsulant is dissipated directly on the thermal pad, bypassing the die and the DAL. In phosphor converted LED package, the die and the DAL experience additional heat flow caused by the phosphor losses in particles located close to the die crystal. An important remark is that this heat flow is significantly delayed due to low thermal diffusivity of silicone/phosphor composite. Figure 9a confirms the "slow" impact of the delayed phosphor heat flow on the pn junction temperature transient. These results allow us to conclude that the phosphor heat generation have no significant impact on the initial thermal transient, yet it partially increases the thermal flux flow through the die and the DAL during steady state.

It should be noted, that the dome phosphor-related heat sources $P_{hD}$ significantly impact the "tails" of the SFs, in Figure 10a. Indeed, as shown above, the phosphor dome heat generation $P_{hD}$ produces an additional delayed thermal flow that affects the slow part of the $T_j$ response. Based on this observation, we conclude that the presence of the distributed heat sources in the dome leads to

the distortion of the regions of SFs correspondent to structures with relatively "slow" time constants (e.g. PCB and the further thermal path). Therefore, application of such techniques as transient dual interface measurements [67] may be limited in cases of LEDs with phosphor light conversion.

The analysis presented in this chapter shows that the value of total thermal resistance of an LED (as derived from the thermal transient measurement) decreases when the fraction of the secondary heat sources increases. This decrease of thermal resistance is associated with the fact that a significant part of the thermal power from secondary heat sources is distributed over the LED package volume. Thus, a part of the total heat losses used to calculate the LED's thermal resistance value bypasses the pn junction, the die, and the DAL. This effectively decreases the junction temperate comparing to the case when all the thermal power is generated only by the pn junction.

### 3.1.4. Die and DAL Thermal Characterization

We notice a "scaling" effect of the SFs presented in Figure 10a: the higher the $P'_h/P_h$ fraction the lower the $R_{th}$ and the higher the $C_{th}$ values. At the same time, the thermal properties of the main heat path remain constant in all the numerical experiments. Moreover, we have noticed the characteristic dependence of $R_{th\_J2T\_err}$ on $P'_h/P_h$, and assumed that the initial heat propagation through the die and DAL is not dependent on the secondary heat sources $P'_h$. In order to verify this hypothesis, we use $P_{hJ}$ instead of $P_h$ as the power step for thermal transient analysis. The resulted new SFs are presented in Figure 11.

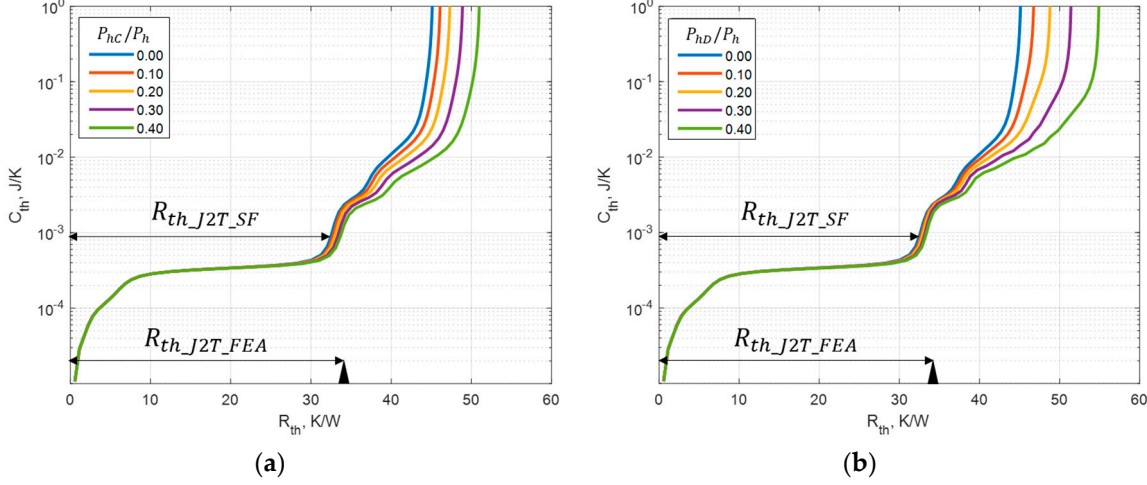

**Figure 11.** FEA derived SFs, when $P_{hJ}$ power step is used for thermal transient analysis. Junction to thermal pad resistances derived by the SF analysis and a reference FEA are indicated: (**a**) when only trapped light losses are considered as the secondary heat source; and (**b**) when only phosphor dome light conversion losses are considered as the secondary heat source.

We observe a cancelation of the "scaling" effect up to the thermal pad step regardless of the nature of the secondary heat source. The derived $R_{th\_J2T\_SF}$ values are almost identical to the reference $R_{th\_J2T\_FEA}$. The $R_{th\_J2T\_err}$ error is reduced to 5% and does not depend on the $P'_h/P_h$ ratio anymore. Thus, we conclude that our hypothesis is valid: the distributed secondary heat sources are not influencing the initial heat propagation though the die and the DAL. We conclude that the die and the DAL thermal properties can be extracted from thermal transient measurements quite accurately if the secondary heat sources are subtracted from the total thermal power.

The properties of the other elements in the thermal path and the total thermal resistance of the LED become distorted if one corrects SF for the power from the secondary heat sources. This can be seen as a spread of the SFs shapes after the thermal pad step. These results show that this method of LED thermal transient analysis can be applied only to characterize the thermal properties of the die and the DAL.

### 3.2. Experimental Verification

We evaluate the thermal power dissipated at the junction $P_{hJ}$ and the secondary heat losses $P'_h$ for each physical LED configuration. To evaluate these parameters we measure $P_{el}$ and $P_{opt}$ and perform initial transient analysis by Equations (12) and (14) in the range of 50–400 μ sec. The obtained thermal power distributions are presented in Table 3.

**Table 3.** Thermal power distribution.

| LED Sample | Dome Configuration | $P_{el}$ (W) | $P_{opt}$ (W) | $P_h$ (W) | $P_{hJ}$ (W) | $P'_h/P_h$ |
|---|---|---|---|---|---|---|
| 1A | Bare chip | 0.444 | 0.189 | 0.255 | 0.250 | 0.02 |
| 1B | Flat silicone | 0.445 | 0.122 | 0.323 | 0.229 | 0.29 |
| 2A | Bare chip | 0.441 | 0.199 | 0.242 | 0.221 | 0.09 |
| 2B | Rim dome | 0.442 | 0.159 | 0.283 | 0.203 | 0.28 |
| 2C | Phosphor top | 0.441 | 0.091 | 0.350 | 0.233 | 0.33 |

First, we analyze the log time derivative of the measured junction temperature $T_J$ response. The measurement data are presented in Figure 12.

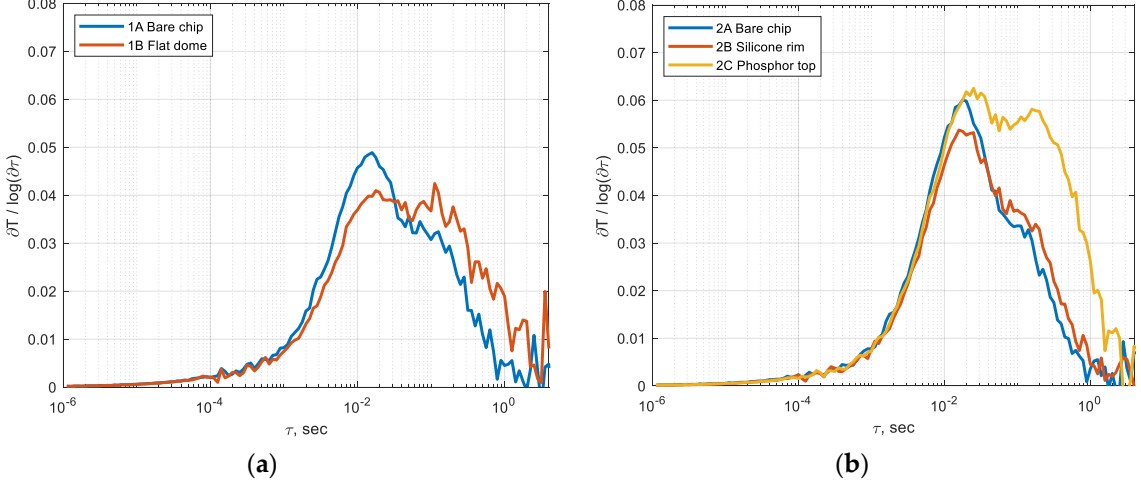

(a)                                                    (b)

**Figure 12.** Experimentally derived log time derivative of $T_j$ response: (**a**) 1st LED; and (**b**) 2nd LED.

Despite a significant variation of the measured total thermal power $P_h$ values, we observe an expected small difference for the initial thermal transient response between both LEDs in our set of experiments. Indeed, alterations of the dome configuration mainly change the power of the secondary heat sources $P'_h$ and weakly affect the pn junction heat power $P_{hJ}$. The initial $T_J$ increase rate is dependent on $P_{hJ}$ according to Equation (12). Moreover, we confirm the predicted impact of the phosphor heat generation on the slow time constants (Figure 12b). Additionally, we notice that the measurement data noise patterns up to 10 msec look similar. This may be an evidence for a presence of a systematic error that is most likely caused by the measurement equipment or a raw data processing algorithms.

The results of the thermal transient testing of the 1st LED are presented below. We employ the experimentally derived $P_h$ and $P_{hJ}$ values as power inputs for the transient analysis.

The comparison of the SFs presented in Figure 13a exhibits the "scaling" effect demonstrated with the FEA modeling. The 1B flat silicone dome configuration has a slightly higher fraction of the secondary thermal losses. Our reasoning for this is:

- The silicone dome enhances the light extraction from the chip into the encapsulant;
- The enhanced light extraction from the chip leads to a reduction of $P_{hJ}$ (Table 3); and

- The light is trapped in silicone dome due to TIR and is absorbed on the reflector surfaces, this effect increases $P'_h$.

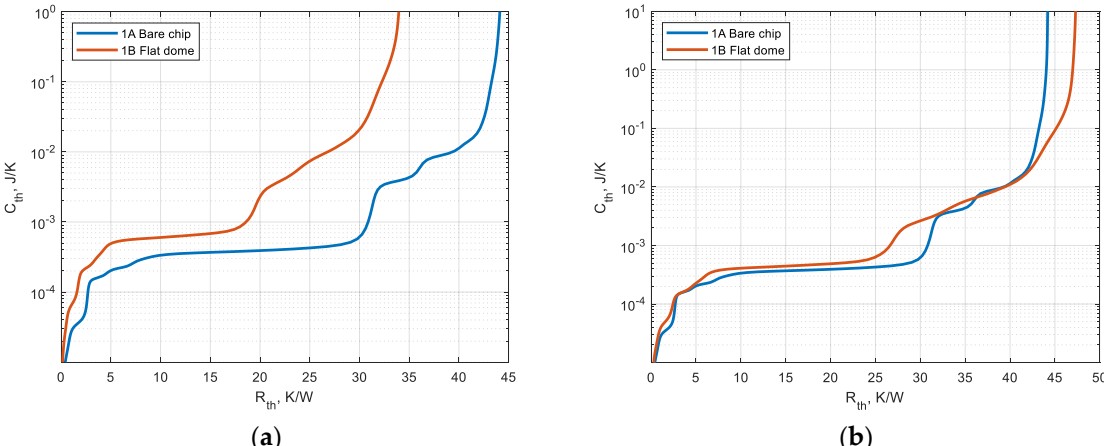

(a)                    (b)

**Figure 13.** SFs of 1st LED: (**a**) Total thermal power $P_h$ is used for the transient analysis; and (**b**) only junction thermal power $P_{hJ}$ is used for the transient analysis.

Now we analyze the SFs when $P_{hJ}$ is used for the transient analysis. The resulting SFs are presented in Figure 13b. The SFs are aligned up to the 25 K/W. The scaling effect is significantly reduced. Yet, considerable discrepancy is observable. The SF of the 1B configuration is significantly "smoother" compared to the bare chip one. We have shown in our previous works [41,43] that this effect is related to the heat storage in the dome encapsulating material.

Analyzed SFs of the 2nd LED are presented in Figure 14. Again, we observe the "scaling" effect when the total thermal power $P_h$ is used for thermal transient analysis (Figure 14a): the higher the $P'_h / P_h$ ratio the more distorted the initial SF regions representing the sapphire chip and the DAL. Like the dome of 1B configuration, the rim of the 2B configuration enhances the light extraction from the crystal, and it increases the fraction of the secondary thermal losses $P'_h / P_h$. Introduction of the phosphor top layer increases the $P'_h / P_h$ ratio even more.

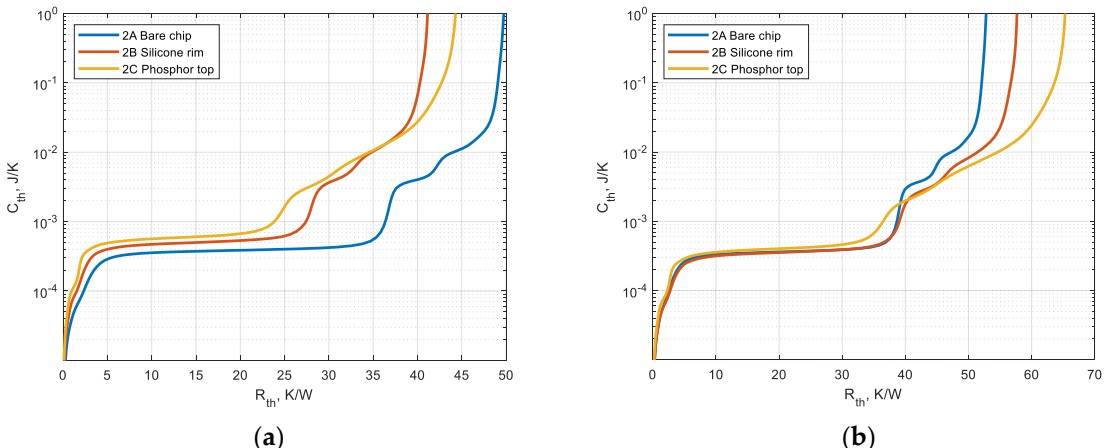

(a)                    (b)

**Figure 14.** SFs of 2nd LED: (**a**) Total thermal power $P_h$ is used for the transient analysis; and (**b**) only junction thermal power $P_{hJ}$ is used for the transient analysis.

Figure 14b demonstrates that usage of the $P_{hJ}$ power for transient analysis suppresses the scaling effect. The resulting 2A and 2B configurations' SFs perfectly overlay each other up to the thermal pad step. The silicone rim does not create a significant parallel capacitive thermal heat path, unlike

the full silicone dome, thus, 2B SF is not blurred. Yet, the 2C configuration SF is blurred due to the secondary heat path created by the phosphor top layer. The derived sapphire chip thermal capacitance is almost identical for all three SFs. The SF of the 2C configuration has a significantly expanded tail compared to the pure silicone dome configuration 2B. This is caused by the phosphor heat generation impact demonstrated with the numerical simulations.

The error of the junction to thermal pad thermal resistance evaluation with thermal transient measurements without the correction for the secondary heat sources is 35% for configuration 1B, 30% for configuration 2B and 37% for 2C. The error is proportional to $P'_h/P_h$ ratio. If the pn junction is completely encapsulated into the dome the error increases by approximately 5% due to the SF distortion caused by heat storage in the dome.

## 4. Discussion

We have numerically and experimentally proven the applicability of the proposed secondary heat sources separation method. Moreover, we have demonstrated that, firstly, the die and the DAL thermal resistance can be accurately derived from a SF only if the correction for the secondary heat sources is done and the thermal power dissipated exclusively by the pn junction is used for the transient analysis. Secondly, we have shown that the trapped light and the phosphor heat generation have no significant influence on the initial transient. Thirdly, we have demonstrated that the phosphor heat generation is affecting the slow time constant region of the transient spectrum and, as a result, it changes the parts of SFs correspondent to such bulky elements as the LED package and assembly elements (e.g. PCB).

Next, we analyze the topologies of various thermal models and calibration procedures used by other authors based on the obtained knowledge. Works [22] and [68] use a bidirectional thermal resistance network to model the thermal behavior of LEDs. This bidirectional model is presented in Figure 15a. It contains two power sources modeling the active region and the dome heat generation. The nodes connected by thermal resistances represent the phosphor dome, the junction and the case of an LED. To derive the model, the authors compare and analyze SFs of bare chip, silicone and silicone/phosphor composite dome packages of the same LED design. The SFs have a noticeable decrease of the cumulative thermal resistance and an increase of the cumulative thermal capacitance of the die and DAL regions evidencing the "scaling" effect demonstrated in Figure 10a. The authors explain the junction to ambient thermal resistances decrease by the enhancement of the heat conduction through the dome towards ambient (e.g., a decrease of $R_{th\_D2J}$ and $R_{th\_D2A}$). Nevertheless, as shown in other scientific works, the steady-state convection and radiation heat transfer from the top of an MP LED package is negligible comparing to the conduction heat transfer via the main heat path [43,44,69] ($R_{th\_D2A} \gg R_{th\_J2C}$). Thus, the bidirectional model proposed by the authors [22] and [68] does not provide a physics-based explanation of the observed phenomena.

Juntunen et al. also developed a thermal resistance network model that considers the secondary heat sources [13]. The model is shown in Figure 15b. The model contains a thermal pad node correspondent to the temperature of the top of the substrate on which an LED die is mounted. The authors consider a one-dimensional heat path but do not separate the junction and the phosphor heat sources in their model. This may not accurately capture all relevant physical phenomena. The SF data presented in their paper also exhibits the "scaling" effect. Yet, they do not consider distributed secondary thermal sources. They explain this effect only by an additional parallel heat path (represented as $R_{th\_shunt}$) formed by the dome that shunts the die and the DAL thermal resistance $R_{th\_J2T}$.

Based on the analysis of the SFs we propose a thermal resistance network model that eliminates the above mentioned flaws. The topology of the model is shown in Figure 15c. It contains three power sources responsible for heat generation in the pn junction, the dome and on the reflective cup surface. The dome heat source is connected both to the pn junction and the thermal pad. The trapped light heat source is located at the thermal pad node. The secondary heat sources are spread over the package according to their actual placement. Thus, a significant part of the heat generated by them is dissipated through the thermal pad bypassing the die and the DAL. This better explains the main reason of the

total $R_{th}$ decrease for the LEDs with secondary heat sources rather than assumed increase of the heat flow through the top of the dome, or the shunting heat path for the die and the DAL. We propose a possible extraction procedure for the parameters of this compact thermal model in Appendix C.

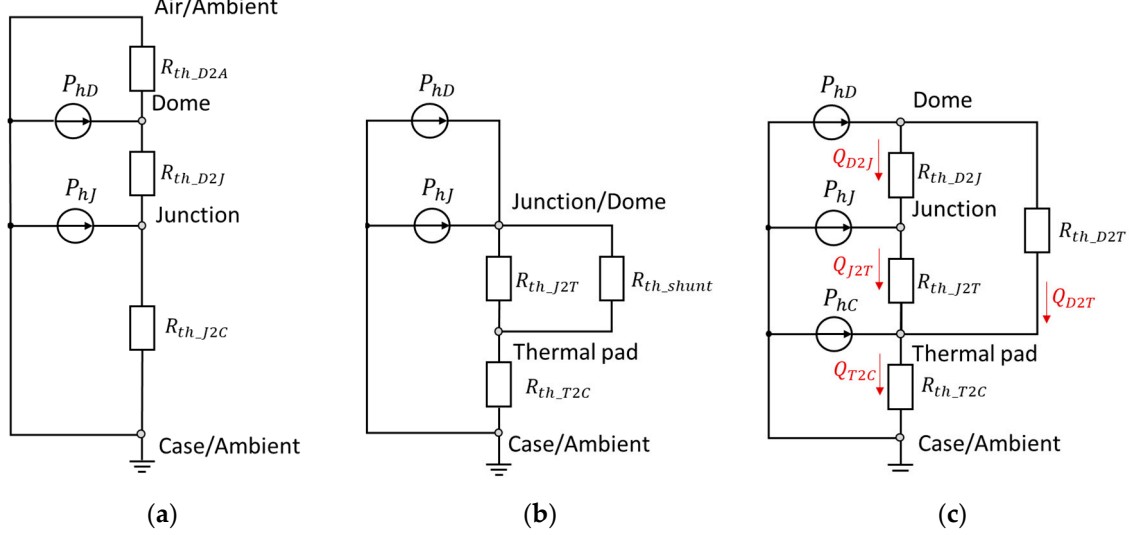

**Figure 15.** LED package thermal resistance models. (**a**) Bidirectional model presented in works [22,68]; (**b**) the model presented in work of Juntunen et al. [13]; and (**c**) the model proposed by us. Heat flows are indicated with red.

The discussed impact of the phosphor secondary heat source on the "tails" of SFs is noticed in the experimental and numerical simulations data of works [13,15,22]. Like demonstrated in Figure 10a, the presence of phosphor losses leads to an extension of the SFs' thermal pad and PCB regions' SF images. As discussed earlier, this effect is caused by the relatively low thermal diffusivity of silicone/phosphor composite and associated alteration of the slow time constant spectrum, as demonstrated in Figure 9a. This knowledge coupled with the demonstrated procedure of heat source separation could enhance the accuracy of the CTM calibration procedure demonstrated in the work of Bornoff et al. [7] and similar ones.

## 5. Conclusions

We have shown that the secondary heat sources have a significant impact on the accuracy of the thermal transient analysis results, in particular, they distort the die and the DAL regions of SF causing a "scaling" effect. We have numerically modeled and experimentally confirmed the distortion of the SF's die and the DAL regions associated with the presence of secondary heat sources. We have experimentally confirmed that the related error of the die and the DAL thermal resistance evaluation can be as high as 35%. Moreover, our estimations indicate that this error can reach up to 50% for LEDs with phosphor light conversion. We have shown that this error is proportional to the fraction of the secondary heat sources in the total power dissipation.

We have proposed a novel method of separation of the main and the secondary heat sources based on the thermal transient analysis and radiant flux measurements. The method is confirmed both numerically and experimentally. The method can significantly increase the accuracy of calibration procedures for 3D FEA thermal models.

We have proposed a technique that can be used to determine actual thermal properties of the die and the DAL. The technique enables suppression of the distortion effects in SFs caused by the secondary heat sources. Using the new SF analysis insights obtained with this technique, we have analyzed current thermal resistor models for multiple heat source LEDs. We notice that these models

are partially derived using incorrect SF interpretations. Thus, we propose a new model topology that enables physically accurate thermal modeling of multiple heat source LEDs.

The characteristic impact of the heat generation by phosphor particles on the transient response was demonstrated. We have found that the heat caused by the phosphor light conversion losses affects "slow" thermal time constants. These time constants are often related to the heat transfer through the PCB. This is an important observation for enhancing calibration accuracy of 3D FEA thermal models of phosphor converted LEDs against thermal transient results.

We have revised and improved the analytical solution for the initial transient response, by considering a case of bilateral heat propagation. Our approach increases the accuracy of the junction thermal power estimation up to 15% for LEDs with phosphor light conversion and up to 5% for the LEDs with clear silicone encapsulant.

In a general case, only the total power of the secondary heat sources can be derived with the proposed secondary heat sources estimation method. It appeared not yet possible to separate the phosphor heat generation and the reflection losses. The sensitivity analysis of the proposed method to the accuracy of the k-factor calibration and the measurement noise, as well as additional cross-verification with other methods of the secondary heat source evaluations should be performed.

Moreover, the heat generation profile in a silicone/phosphor composite dome is not uniform. It depends on the phosphor particle spatial distribution around the die, the dome shape and the reflection cup geometry. Thus, the secondary heat sources power density distribution within an LED package is difficult to evaluate. There is no technique yet to estimate the part of the heat dissipated by the phosphor particles that bypasses the die and DAL. As a result, the proposed power separation method still cannot guarantee a precise calibration if used alone without FEA. Thus, we see room for future research.

**Author Contributions:** Methodology, software, validation, writing—original draft preparation: A.A.; conceptualization: A.A. and G.O.; supervision, writing—review and editing: J.-P.L. and G.M.; project administration, and funding acquisition: G.M.

**Funding:** This research has received funding from the European Union's Horizon 2020 research and innovation program through the H2020 ECSEL project Delphi4LED (grant agreement number: 692465) (2016–2019). Co-financing of the Delphi4LED project by the national R&D funding organization of the participating countries. Additional information is available on: www.DELPHI4LED.eu.

**Acknowledgments:** Support from Delphi4LED project partners, especially from Robin Bornoff (Mentor), Andras Poppe (Mentor), Marta Rencz (Mentor) and Gabor Farkas (Mentor) is acknowledged.

**Conflicts of Interest:** The authors declare no conflict of interest.

## Nomenclature

| | |
|---|---|
| CTM | Compact thermal model |
| DAL | Die attach layer |
| DUT | Device Under Test |
| FEA | Finite Element Analysis |
| LED | Light-emitting diode |
| PCB | Printed circuit board |
| MCPCB | Metal core PCB |
| MP | Mid-power |
| SF | Structure function |
| TIR | Total internal reflection |
| *EQE* | External quantum efficiency |
| *IQE* | Internal quantum efficiency |
| *I* | Current |
| *a* | Nondimensional parameter characterizing composite particles |
| *f* | Phosphor particles volume fraction |
| $\Lambda$ | Crystal light re-absorption correction coefficient |
| $P_{el}$ | Electric power |

| | |
|---|---|
| $P_{opt}$ | Radiant flux |
| $P_h$ | Total thermal power |
| $P'_h$ | Combined secondary heat sources thermal power |
| $P_{hJ}$ | Thermal power dissipated in a pn junction |
| $P_{h\_re}$ | Thermal power dissipated in a crystal during the trapped light re-entries |
| $P_{hC}$ | Thermal power dissipated on the cup reflector surface |
| $P_{hD}$ | Thermal power dissipated in the dome volume |
| $C_{th}$ | Thermal capacitance |
| $R_{th\_y}$ | Thermal resistance |
| $Q_y$ | Heat flow |
| $y$ | suffix made of two capital letters separated by "2". The letters are can be "*J*", "*D*", "*T*", and "*C*", denoting pn junction, and the LEDs' dome, top of the thermal pad and case. The suffix denotes elements connecting two entities. The suffix can be followed by "FEA" which designates value derived directly from finite element analysis, "SF" which designates value derived from a structure function and "err" which designated the relative error of the value. |
| $S_{pn}$ | pn junction area |
| $S_{cup}$ | Reflector cup area |
| $S_{cry}$ | Crystal chip area |
| $N_{cup}$ | Average number of the photons bouncing off the cup walls |
| $N_{cry}$ | Average number of crystal re-entries by photons |
| $\tau_z$ | Time |
| $T_z$ | Temperature |
| $\rho_z$ | Density |
| $k_z$ | Thermal conductivity |
| $C_z$ | Specific heat |
| | Spatial coordinate |
| $q_z$ | Heat flux |
| $H_z$ | Height |
| $R_z$ | Radius |
| $V_z$ | Volume |
| $z$ | suffix that can be "*a*", "*j*", "*d*", "*s*", "*sil*", "*pho*", and "*sap*" denoting ambient, pn junction, LEDs' dome, pn junction substrate, silicone, phosphor, and sapphire materials, respectively |
| $K_{uni\_lat}$ | Unilateral heat propagation initial transient constant |
| $K_{bi\_lat}$ | Bilateral heat propagation initial transient constant |
| $LEE$ | Total light extraction efficiency |
| $LEE_{chip}$ | Chip to dome light extraction efficiency |
| $LEE_{dome}$ | Dome to ambient light extraction efficiency |

## Appendix A

This appendix enables an analytical estimation of the reabsorption coefficient $\Lambda$ dependence on the LEDs dome and cup geometries.

The optical losses from the portion of the light when it left the crystal for the first time $P_{h\_opt}$ are equal to:

$$P_{h\_opt} = IQE(I) \cdot LEE_{chip} \cdot (1 - LEE_{dome}) \tag{A1}$$

These losses are dissipated either on the reflector cup surface as $P_{hC}$ or inside of the crystal during light re-entries as $P_{h\_re}$:

$$P_{h\_opt} = P_{hC} + P_{h\_re} \tag{A2}$$

The equation can be rewritten with the coefficient $\Lambda$:

$$P_{hC} = P_{h\_opt} \cdot \Lambda \tag{A3}$$

$$P_{h\_re} = P_{h\_opt} \cdot (1 - \Lambda) \tag{A4}$$

The ratio between $P_{hC}$ and $P_{hJ\_re}$ is determined by the number of the cup reflection events $N_{cup}$, the crystal transitions events $N_{cry}$ and the correspondent probabilities of light absorption during these events. The absorption probabilities are determined by the cup walls reflection coefficient $R_{cup}$ and the crystal transition coefficient $T_{cry}$:

$$\frac{P_{hC}}{P_{h\_re}} = \frac{N_{cup}\left(1 - R_{cup}\right)}{N_{cry}\left(1 - T_{cry}\right)} \tag{A5}$$

We assume that the average number of the photons bouncing off the cup walls and the number of crystal re-entries by the trapped light are proportional to their surface areas exposed to the dome. Thereby, the ratio between these quantities is:

$$\frac{N_{cup}}{N_{cry}} \cong \frac{S_{cup}}{S_{cry}} \tag{A6}$$

where $S_{cup}$ is the area of the dome cup walls, $S_{cry}$ is the area of the sapphire crystal in contact with the encapsulating dome. Now, assuming that the crystal transition probability is approximately equal to the light extraction efficiency of the chip: $T_{cry} \cong LEE_{chip}$, $\Lambda$ coefficient can be determined based on Equations (A3)–(A6):

$$\Lambda \cong \left(1 + \frac{S_{cup}\left(1 - R_{cup}\right)}{S_{cry}\left(1 - LEE_{chip}\right)}\right)^{-1} \tag{A7}$$

## Appendix B

In this appendix, we present an analytical solution for the transient temperature response of the interface between two semi-infinite bodies when a uniform heat source is located at the contact. One of the bodies represents the crystal substrate, the other one corresponds to the dome. Figure A1 illustrates the problem.

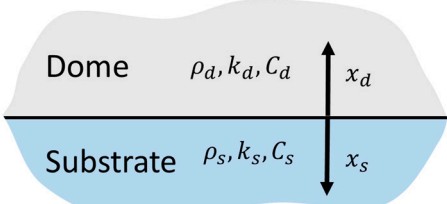

**Figure A1.** Transient heat conduction on the edge of the sapphire crystal and dome. $x_d$ and $x_s$ are the spatial coordinates.

The problem is one-dimensional. At the initial time $\tau = 0$ the bodies have uniform identical temperatures $T_0$. A constant and uniform heat source with power density $q$ is activated at the interface at $\tau = 0$. The heat transfer governing equations are:

$$\rho_d C_d \frac{\partial T_d}{\partial \tau} = k_d \frac{\partial^2 T_d}{\partial x_d^2} \tag{A8}$$

$$\rho_s C_s \frac{\partial T_s}{\partial \tau} = k_s \frac{\partial^2 T_s}{\partial x_s^2} \tag{A9}$$

The initial and the interfacial conditions are:

$$\begin{aligned} T_d(x_d, 0) &= T_0 \\ T_s(x_s, 0) &= T_0 \end{aligned} \tag{A10}$$

$$T_d(0, \tau) = T_s(0, \tau) \tag{A11}$$

We define the heat fluxes $q_d$ and $q_s$ that leave the interface towards the dome and the sapphire, respectively. According to the law of conservation of energy, their sum is equal to $q$. Thus:

$$q_d = -k_d \frac{\partial T_d(0, \tau)}{\partial x_d}\bigg|_{x_d=0} \tag{A12}$$

$$q_s = -k_s \frac{\partial T_s(0, \tau)}{\partial x_s}\bigg|_{x_s=0} \tag{A13}$$

$$q = q_d + q_s \tag{A14}$$

From Equations (A8), (A9), (A11)–(A13), it can be seen that $q_d$ and $q_s$ are constant and they depend only on the thermal properties of the sapphire crystal and the dome. The ratio between the heat fluxes is:

$$\frac{q_d}{q_s} = \sqrt{\frac{k_d \rho_d C_d}{k_s \rho_s C_s}} \tag{A15}$$

Therefore, the bilateral heat propagation problem can be reduced to two unilateral problems with heat fluxes $q_d$ and $q_s$. The heat flux through the sapphire crystal can be expressed as:

$$q_s = \frac{q}{1 + \sqrt{\frac{k_d \rho_d C_d}{k_s \rho_s C_s}}} \tag{A16}$$

Thus, the surface temperature response is equal to:

$$T(t) = T_0 + \frac{2q}{\sqrt{\pi} \left( \sqrt{k_s \rho_s C_s} + \sqrt{k_d \rho_d C_d} \right)} \sqrt{\tau} \tag{A17}$$

## Appendix C

The calculations presented in this appendix show a way to estimate the parameters of our thermal resistance model shown in Figure 15c.

Using simple circuit analysis, we define the dependence of the heat flows $Q_{D2J}, Q_{J2T}, Q_{D2J}$, and $Q_{T2C}$ on the corresponding thermal resistances and the average temperature of the junction $\overline{T}_j$, the average temperature of the upper surface of the thermal pad $\overline{T}_t$, the average case temperature $\overline{T}_c$ and the chosen measure of the dome temperature $T_d$:

$$Q_{D2J} = \frac{T_d - \overline{T}_j}{R_{th\_D2J}} \tag{A18}$$

$$Q_{J2T} = \frac{\overline{T}_j - \overline{T}_t}{R_{th\_J2T}} \tag{A19}$$

$$Q_{D2T} = \frac{T_d - \overline{T}_t}{R_{th\_D2T}} \tag{A20}$$

$$Q_{T2C} = \frac{\overline{T}_t - \overline{T}_c}{R_{th\_T2C}} \tag{A21}$$

The Kirchhoff's current law bounds the heat flows and the considered heat sources:

$$P_{hD} = Q_{D2J} + Q_{D2T} \tag{A22}$$

$$P_{hJ} = Q_{J2T} - Q_{D2J} \tag{A23}$$

$$P_{hC} = Q_{T2C} - Q_{J2T} - Q_{D2T} \tag{A24}$$

The collection of Equations (A18)–(A24) creates an underdetermined system with 15 variables. However, $\overline{T}_j$ and $\overline{T}_c$ can be estimated by a thermal transient measurement and a dual interface material method. We have shown that $R_{th\_J2T}$ can be found from an SF if the junction thermal power $P_{hJ}$ is used for thermal transient analysis instead of the total thermal power $P_h$. A method of estimation of $P_{hJ}$ and the sum of the secondary heat sources $P_{hD}$ and $P_{hC}$ is also described in this paper.

Firstly, we consider an LED without phosphor light conversion layer. In this case, $P_{hD} = 0$ and the dome temperature is not a critical reliability parameter. Moreover, the parallel heat path through the dome formed by $R_{th\_D2J}$ and $R_{th\_D2T}$ has very high characteristic thermal resistance compared to the main thermal path due to low thermal conductivity of silicone [43]. Thus, Equations (A18), (A20), and (A22) associated with the dome node can be taken out of the consideration. $Q_{D2J}$ and $Q_{D2T}$ can be assumed to be equal to zero. Now, the system can be solved and the rest of the model parameters can be determined mathematically.

Secondly, we consider an LED with a phosphor light conversion layer. The phosphor particles encapsulated in the dome suppress multiple TIRs due to light re-emission and light scattering. Thus, we can assume that the trapped light reflection losses $P_{hC}$ are negligible and all the secondary heat losses are primarily due to $P_{hD}$. The temperature of the phosphor dome is not uniform in a general case. More research is required to understand the temperature profile of it [70]. Yet, a number of methods, like infrared imaging, thermal couple measurements,

or spectroscopic approaches, are known for phosphor layer temperature characterization [71–74]. Any of these methods can be chosen to provide a characteristic measure of the dome temperature $T_d$. Part of the heat produced by the phosphor dome dissipates through the pn junction and the sapphire, the other part goes directly to the thermal pad. Thus, the ratio of $R_{th\_D2J}$ and $R_{th\_D2T}$ can be approximated by the reciprocal of the ratio of the crystal and the cup surfaces $S_{cry}$ and $S_{cup}$. Now the considered system of equations becomes determined. Therefore, all the model parameters can be calculated.

The accuracy of phosphor and junction temperature modeling by a thermal resistor network is validated in [22]. Our model can be mathematically transformed into the one used in the reference by network analysis techniques, thus, it can also be considered as validated. Yet, our model provides a physics-based explanation of the thermal behavior of an LED.

The presented method for parameter extraction enables analytical derivation of the models parameters. Nevertheless, such parameters as $R_{th\_D2J}$ and $R_{th\_D2T}$ can only be coarsely estimated by SF and LED package geometry analysis. As discussed earlier in the paper, such parameters can be extracted with a higher accuracy from calibrated thermal finite element models.

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
