# Peer review of "Multiple Heat Source Thermal Modeling and Transient Analysis of LEDs"

_energies, doi:10.3390/en12101860_

Reviewer 1 Report

Please see attached my comments.

Author Response

Word document is uploaded

Reviewer 2 Report

This paper aims to investigate the inaccuracies of thermal transient measurements interpretation associated with the secondary heat sources related to the dome trapped light and phosphor light conversion losses. The paper deals with an interesting subject precious of investigation. I consider that the manuscript can be published after some revision.

Some remarks that could improve the paper are:

-Whereas the introduction, the model development, and explanation are well done, the result descriptions should be better presented. Please avoid using repeated sentences.  

-Please give more details in the experimental part which are inadequately explained. The authors should clearly indicate the contribution and limitations of their work.

-Dealing with the results, from the reviewer point of view, the validation part is not sufficiently interpreted. The results should be further elaborated to show how they could be used for the same applications.

-The comparison part is missed. The authors should include in the paper some comparison with experimental and numerical data of other authors. So, your results contribution in comparison to the references should be strengthened.

-The authors only show the sensitivity of the model to parameters. Generally, the comparisons between the simulations must be deeper. The conclusion could be itemized based on the significance of the work.

-An updated and complete literature review must be conducted and at the end of that please clearly state the novelty of your research. 

Author Response

Notes are in the attached file

Reviewer 3 Report

This paper introduces and analyze a methodology of calibration of thermal models for multiple heat source LEDs characterization. In my opinion, this paper is interesting but there are some revisions that I suggest to address:

-Sections 1 and 2 (i.e. Introduction and Materials and Methods) are okay, but they would benefit from some additional references to help improve the academic rigour and also the general readability for readers.

-Methods need more explanations and specifications to enable the reader to follow. 

-I suggest adding a flow-chart representing the main step of the research and providing more information about the methodology.

-The plots are not sufficiently analysed.

-The quality of the communication needs to be improved. This partly relates to improving the English, but more to the explanation of issues to the reader.

Author Response

The notes are in the attached flie

Reviewer 4 Report

Dear Author,

the paper is very voluminous. The consideration of multiple heat sources is implemented in several scientific works, but it is not really described and investigated in a proper way. Thus, your work is filling a gap by describing the methodology of the use of multiple heat sources by the thermal analysis of phosphor converted LED systems.

As already mentioned, the paper is very voluminous, and sometimes the content is running out of focus. I would ask for a rework of some parts of your work, as well as a focusing just on one example, here the MP-LED System. As experiments were only carried out on the MP-LED System. The COB design is only investigated in the FEA, but no Transient Analysis is done. As already published by (F. Profumo, A. Tenconi, S. Facelli, and B. Passerini, “Implementation and validation of a new thermal model for analysis, design, and characterization of multichip power electronics devices,” IEEE Trans. Ind. Appl., vol. 35, no. 3, pp. 663–669, 1999); (A. Poppe and A. Szalai, “Practical Aspects of Implementation of a Multi-domain LED Model,” in 30th SEMI-THERM Symposium, 2014.); (L. Mitterhuber et al., “Thermal transient measurement and modelling of a power cycled flip-chip LED module,” Microelectron. Reliab., 2017.) They investigated the multi-chip behavior and an assumption of summing up the power of chips, without thinking of cross-sensitivities is ok, but it has to be seen critically in terms of reality checks.

In the introduction the author mentioned, that there is less research in the field of phosphor conversion as a function of temperature. But, a short literature research showed a quite different view. Please rewrite this statement!

M. Dal Lago et al., “Phosphors for LED-based light sources: Thermal properties and reliability issues,” Microelectron. Reliab., vol. 52, no. 9–10, pp. 2164–2167, 2012.

M. Arik, C. a. Becker, S. E. Weaver, and J. Petroski, “Thermal Management of LEDs: Package to System,” Proc. SPIE, vol. 5187, pp. 64–75, 2004.

E. Liu, A. Hanss, M. Schmid, and G. Elger, “The influence of phosphor layer and sidecoating on the thermal performance and the structure function of modern waver level high power LEDs,” THERMINIC 2015 - 21st Int. Work. Therm. Investig. ICs Syst., vol. 2015, no. October, pp. 2–5, 2016.

P. Fulmek et al., “The impact of the thermal conductivities of the color conversion elements of phosphor converted LEDs under different current driving schemes,” J. Lumin., vol. 169, pp. 559–568, 2016.

E. Juntunen, O. Tapaninen, A. Sitomaniemi, and V. Heikkinen, “Effect of Phosphor Encapsulant on the Thermal Resistance of a High-Power COB LED Module,” IEEE Trans. Components, Packag. Manuf. Technol., vol. 3, no. 7, pp. 1148–1154, Jul. 2013.

The derivation of the influence of a multi heating source system –equation 13-21- are confusing. A lot of new abbreviation were used and not described well. From my point of view, this should be given in the APPENDIX!

In the paper FEA simulation were done, but it is not mentioned which software is used nor which boundary or initial conditions are used. Some simulation pictures with a temperature distribution could visualized the heating behavior of the system. Also to point out clearly where the monitoring points were placed for doing the Transient Temperature Analyses.

In the experimental part of this paper, only the SF is shown. But the author mentioned several time, that multi heat sources should be taken into account, which can be seen in the thermal transient! It was already done for the simulation, it should be also done for the experiment.

Concerning the experiment, the setup should be mentioned in some words!

In the first chapters the authors introduced 2 systems (MP LEDs and COB systems). At the end, only MP LEDs were investigated. From my point of you, a focus on only one system – here the MP LEDs- should be done. A multi chip system needs more investigation, as already mentioned before!

Nomenclature: A huge amount of abbreviation are listed, but there are some missing, which are used in the paper: H, Λ,α, V, f etc.…. are missing in the list.

Sometimes the abbreviations are described in the text, sometimes not. Inconsistencies!

Line 114: The characteristic features and the geometrical size of these LED packages allow us to determine the range of the FEA thermal model parameters for further analysis.

From your point of view: what are characteristic features? And how can you determine the range of FEA?

Figure 3 and line 161-164: what is the methodology used, to correlate the geometry with regions of the SF? And are you sure, that the PCB is located in this region you are mentioned there? The Figure 3 is not correlated with any sketch before. The sketches have no PCB and no heatsink in.

Line 167: The heat flow to the thermally inactive dome creates an extra heat path during the thermal transient processes.

What does inactive mean: the dome or also called glob-top is treated like air?

Line 208: error! Reference source not found.

Table 1. C= J/gK -- K is missing

Line 218: describe the boundary conditions in more detail

Line 254: 50°C

Equation 11: a multiplication sign is missing

Equation 18: why do you use now a function of the current f(I) ?

Equation 21: why is LEEdome in this equation.

Equation 26: H²sap – whats that?

Equation 29-- is equation 27 fulfilled?

Line 383: Error! Reference source not found.

Line 383: Thus, we conclude that the heat propagation into the silicone dome is always sufficiently on dimensional while the heat has not reached DAL via Sapphire substrate for a typical LED.

Not clear!

Line 402:

 Zhang et al. [41] have fitted the model of the equation to experimental measurements of the typical silicone/phosphor composite used in LEDs. They used and Ce3+ doped YAG (?3??5?12) phosphor particles of 13.0 ± 2.0 μm diameter encapsulated in high optical transparency silicone. They achieved an excellent agreement with the experimental results for high volume concentrations ? from 20% to 40%. It was found that ? is 0.004 and ?? is 0.032 μm.

? is ?

Line 434: … estimated results.

Which kind of results, where are they?

Line 435: what kind of value is 0.03?

Figure 8: where can I see the 15%. Please sign in.

Line 463:  … blue LEDs are driven in the droop regime when the IQE is approx.. 70%

Marking in the plot for a better visibility!

Line 469: …. PhD can reach up to 50%...

Why do you now? Which kind of equation to you use?

Figure 10: only the solid line are described in the legend, not the dashed line. The difference between solid and dashed is hardly to see!

Table 2: to show the difference between the bilateral and unilateral analytical models, a plot would be fine (1µs-400µs).

Figure 11 b) inconsistency in naming: Dome trapped light = Phc, Phosphor light conversion PhD . Why is there a 5% deviation?

Line 504: as a superposition…

This should be shown!

Line 537: to evaluate these parameters we measure Pel and Popt

Popt is missing in Table 3

Line 604: ..we question the accuracy of dual interface….

This is not proved or discussed before. This claim could be disproved with adequate simulation!

Figure 16: which values are used for the paper?

Author Response

The reply is attached as a Word file

Reviewer 5 Report

The article is very interesting and presents a novel methodology for thermal analysis. I think it would be very appropriate for publication in the magazine with minor revisions.

It is recommended to promote the Bibliography of the introduction with similar current studies to increase the interest of the uninitiated reader and to enhance the line of research developed. There are only 8 references of the 50 of the article.

The methodology used is of great scientific value because of the novelties presented on traditional methods. Although perhaps a greater simplification of them would be advisable. It is excessively long and goes into details that could be dispensable

It would be very interesting in this article to compare the obtained results, at different contour temperatures, by means of the theoretical thermal simulation of the proposed 3D models or using some specific simulation software such as the ANSYS FLUENT to be able to contrast the theoretical results with the practical results Real validating the results with a thermal imaging camera.

It is recommended to review the conclusions to make them better understandable to the reader explaining more precisely the novelty of the method and the results obtained. As well as the future lines of applied research.

In general, we consider that the article may be published with slight modifications

Author Response

The responce is attached as a Word file

Round  2

Reviewer 3 Report

The paper has been improved. I think the revised version is now acceptable.

Author Response

Dear Reviewer,

Thank you for the update.

As I understood, there are no extra comments. Yet, it looks like there was a bug in the system and I have received a request to response them.

Prese correct me if I am wrong.

Best regards,

Anton

Reviewer 4 Report

The author has implemented a large part of the comments and suggestions. In general, it should be noted that the text was formulated very lengthy. Many papers are described in detail, especially is often referred to the already achieved own results. A short and concise summary would be desirable here.

As already noted in the first review report, when describing the characteristic structures of the SF, it should be added how to relate to the geometry of the system (empirical values?)-Figure2.

If IQE is dependent to I, then please note it already in function 8,9,27…IQE(I)

Figure 6: why is there the COB LED compared? The focus lies on the MP!?

Line 528: use the abbreviation of seconds – s!

Author Response

The author has implemented a large part of the comments and suggestions. In general, it should be noted that the text was formulated very lengthy. Many papers are described in detail, especially is often referred to the already achieved own results. A short and concise summary would be desirable here.

èWe recognize that the paper has become lengthy, partly also due to the questions asked in the first review round. We have deleted some extra parts referring to our old results that does not contribute much to the paper now (e.g. lines 134-146).  Yet, we hesitate to condense the paper more as this may conflict with questions and comments by other reviewers made before and would require further iterations. Nonetheless, we believe that the abstract, the final section of the introduction and the conclusions give a fairly concise summary.

As already noted in the first review report, when describing the characteristic structures of the SF, it should be added how to relate to the geometry of the system (empirical values?)-Figure2.

èWe excuse for misunderstanding this point during the first review round. A more detailed description binding the thermal path elements demonstrated in Figure 1 with Figure 2 SF is given now.  A differential SF is plotted together with the cumulative one to illustrate in details how we relate the  to the SF. We do not concentrate on the other thermal path elements since they are not directly considered in our analysis.

If IQE is dependent to I, then please note it already in function 8,9,27…IQE(I)

èdone

Figure 6: why is there the COB LED compared? The focus lies on the MP!?

èThe legend is corrected. The case similar to COB was left to demonstrate the impact of low Scup/Scry ratio that can also be characteristic for MP LEDs if the packages size is small. Yet, we have forgot to remove COB mention from the legend. We have made new plot for  Scup/Scry equal to  7 and 2 instead of 7.02 and 1.87.

Line 528: use the abbreviation of seconds – s!

èdone. The abbreviation is used.  Nonetheless, we use “sec” instead of ”s” since in all the other places we used “sec” as well.